# Assessment of STAT5 as a potential therapy target in enzalutamide-resistant prostate cancer

Holger H. H. Erb[1], Julia Bodenbender[2], Florian Handle[3], Tamara Diehl[2], Lukas Donix[1,4], Igor Tsaur[2], Martin Gleave[5], Axel Haferkamp[2], Johannes Huber[1], Susanne Fuessel[1], Eva Juengel[2☯], Zoran Culig[6☯], Christian Thomas[1☯]*

1 Department of Urology, Technische Universität Dresden, Dresden, Germany, 2 Department of Urology and Pediatric Urology, University Medical Center Mainz, Mainz, Germany, 3 Molecular Endocrinology Laboratory, Department of Cellular and Molecular Medicine, KU Leuven, Leuven, Belgium, 4 National Center for Tumor Diseases (NCT), Dresden, Germany, 5 The Vancouver Prostate Centre, University of British Columbia, Vancouver, Canada, 6 Experimental Urology, Department of Urology, University of Innsbruck, Innsbruck, Austria

☯ These authors contributed equally to this work.
* christian.thomas@uniklinikum-dresden.de

**Data Availability Statement:** All relevant data are within the manuscript and its Supporting Information files.

## Abstract

Despite enzalutamide's efficacy in delaying the progression of metastatic castration-resistant prostate cancer (CRPC), resistance to this anti-androgen inevitably occurs. Several studies have revealed that the signal transducer and activator of transcription (STAT) 5 plays a role in tumour progression and development of drug resistance such as enzalutamide. Data mining revealed heterogeneous expression of STAT5 in enzalutamide-treated mCRPC patients and enzalutamide-resistant prostate cancer (PCa). Isobologram analysis revealed that the STAT5 inhibitor pimozide combined with enzalutamide has? additive and synergistic inhibitory effects on cell viability in the used models. Functional analysis with siRNA-mediated STAT5 knockdown yielded divergent results. The LNCaP-derived cell line MR49F could be resensitised to enzalutamide by siRNA-mediated STAT5b-knock-down. In contrast, neither STAT5a nor STAT5b knockdown resensitised enzalutamide-resistant LAPC4-EnzaR cells to enzalutamide. In conclusion, our results indicate that STAT5 may be a possible target in a subgroup of enzalutamide-resistant PCa. However, based on the data presented here, a general role of STAT5 in enzalutamide-resistance and its potential as a therapeutic target could not be shown.

## Introduction

Prostate cancer (PCa) is the most common cancer in elderly men. According to current numbers of the "Cancer Today" and "Globocan 2018" projects, PCa has the second-highest incidence of all cancers, with 1,276,106 new cases diagnosed (Europe 449,761, USA 234,278) and 358,989 deaths yearly related to PCa worldwide (Europe 41,290, USA 32,686) [1, 2]. Androgen deprivation therapy (ADT) is the primary option for the treatment of locally advanced or

**Funding:** This study was funded by Astellas Pharma. CT and ZC have received the research grant DE72-RG36. This study was also supported by the Open Access Funding by the Publication Fund of the TU Dresden. The funders had no role in study design, data collection and analysis, decision to publish, or preparation of the manuscript.

**Competing interests:** The authors have read the journal's policy and have the following competing interests: This study was funded by Astellas Pharma. CT and ZC have received the research grant DE72-RG36. This does not alter our adherence to PLOS ONE policies on sharing data and materials. There are no patents, products in development or marketed products associated with this research to declare.

metastatic PCa. However, it inevitably leads to disease progression and the development of castration-resistant PCa (CRPC), which is currently incurable [3]. Enzalutamide is a second-generation non-steroidal anti-androgen that has been approved in 2014 for the treatment of CRPC in pre- and post-chemotherapy settings. It inhibits androgen binding to the androgen receptor (AR), receptor translocation into the nucleus, binding to DNA and coregulator recruitment [4]. In patients with non-metastatic CRPC, enzalutamide significantly reduced the risk of metastases by 71% compared with placebo in the PROSPER study [5]. The median metastasis-free survival was 36.6 months with enzalutamide plus ADT versus 14.7 months with placebo (HR, 0.29; 95% CI, 0.24–0.35; P <0.001) [5]. Despite its revolutionary impact on CRPC treatment, enzalutamide is only effective for a certain period before the disease progresses and drug resistance occurs [6, 7]. The resistance mechanisms to androgen deprivation are not fully understood yet. However, several molecular adaptations have already been identified, including AR amplification and mutation, changes in coregulator expression and activation of bypass pathways [6–12]. For example, the AR mutation F877L (alternatively described as F876L based on older genomic builds) was detected in cell-free DNA from the serum of CRPC patients progressing on enzalutamide [13]. In cell models such as MR49F, this mutation caused AR activation by several anti-androgens, including enzalutamide [13–15]. However, a study by Robinson et al. revealed that in patients with mCRPC, of which about a half were pre-treated with enzalutamide, the F877L mutation was not detected and therefore seems to be a rare event [7, 16]. Another identified enzalutamide-resistance mechanism is driven by AR splice variants like the AR-V7 [17]. Li and colleagues could demonstrate that the AR-V7-expressing cell line 22Rv1 was *de novo* resistant to enzalutamide [17, 18]. *In vitro* studies from Bishop and colleagues revealed AR-dependent and -independent mechanisms in enzalutamide-resistant cell models [19]. Puhr et al. and Arora et al. identified the induction of glucocorticoid receptor (GR) expression as a common feature of enzalutamide-resistant tumours in preclinical models as well as patient samples [20, 21]. The groups have proven that the GR confers resistance to anti-androgens by bypassing the AR. A recent study published by Udhane et al. revealed that enzalutamide treatment leads to an AR-mediated activation of the signal transducer and activator of transcription (STAT) 5, thereby, mediating PCa growth. *STAT5* (which refers to two highly related proteins, STAT5a and STAT5b) has been shown to play a pivotal role in the progression of PCa [22–25]. STAT5 expression in human PCa tissue correlates with high Gleason grades and predicts early disease recurrence after initial treatment with radical prostatectomy [26, 27]. PCa xenograft studies demonstrated that STAT5 plays a crucial role in tumour initiation and progression and that high expression of STAT5 has been linked to a mesenchymal phenotype [28, 29]. Thomas and colleagues reported that STAT5 protein expression is increased in human PCa during androgen-deprivation [28]. STAT5 increases the transcriptional activity of the AR by influencing AR protein stability in PCa cells *in vivo* and *in vitro* [12, 28]. This finding is of significant interest as the AR signalling pathway remains active in CRPC despite low levels of circulating androgens [30].

To improve our understanding of the role of STAT5 in enzalutamide resistance, we evaluated its role in cell models and assessed its value as a potential therapeutic target in enzalutamide-resistant PCa *in vitro*.

## Materials and methods

### ChIP-Seq analysis and data mining

For STAT5a and STAT5b expression analysis, the datasets GSE55345, GSE130534, GSE78201, GSE40050, GSE81796, GSE69896, GSE13332, GSE11428, and PRAD_SU2C_2019 were analysed by using Qlucore Omics Explorer v3.5. For AR ChIPSeq analysis we extracted the AR-

binding sites in the cell lines C4-2, LNCaP, VCaP and 22Rv1 from a publicly available GEO dataset (GSE62442). Subsequently, we measured the normalised peak counts in C4-2 cells treated with mibolerone or vehicle in a second publicly available GEO dataset (GSE65066) and visualised the results with the Integrative Genomics Viewer (IGV) [31, 32].

## Cell culture

The human PCa cell line PC3 was obtained from the American Type Culture Collection. C4-2 cells were provided by Prof. Thalmann (University of Berne, Switzerland) [33]. The enzaluta-mide-resistant LNCaP sub cell line MR49F cells were provided by Dr. Gleave [19, 34]. Castra-tion-resistance and enzalutamide-resistance was established by enzalutamide treatment of xenograft mouse models. To eliminate the possibility that the changes observed were due to the mouse microenvironment we choose the CRPC model C4-2 as enzalutamide sensitive model created by passaging twice through mice similar to the MR49F cells (S1A and S1B Fig). The cell lines LAPC4-CTRL, LAPC4-EnzaR, LNCaPabl-CTRL, LNCaPabl-EnzaR, DuCaP-CTRL, and DuCaP-EnzaR were provided by Prof. Culig (Medical University of Innsbruck, Austria) and enzalutamide resistance has been developed by the dose escalation method (S1C and S1D Fig) as described by Hoefer et al. [35]. Culture media used for the cell lines are listed in the S1 Table. All cells were maintained at 37˚C in 5% $CO_2$. Enzalutamide-resistance has been confirmed by the increase of the $IC_{50}$ in cell viability (S1B and S1D Fig). Mycoplasma testing was performed regularly using the Mycoalert Detection Assay (Lonza). Cell line authentication was performed yearly by STR profiling.

## Drug treatment

Enzalutamide (Astellas Pharma, 3343, Lot Number: RS-8BK0189-4) and pimozide (Sigma Aldrich, P1793-500MG, Lot Number: SLBL8929V) were dissolved in DMSO as a 100 mM stock solution and stored as aliquots at -80˚C. Before performing experiments, enzalutamide and pimozide were diluted in the cell culture medium. Isobologram experiments were carried out in the presence of 0.001 to 100 μM enzalutamide as introduced by Loewe et al. [36]. All other experiments were performed in the presence or absence of 10 μM enzalutamide. Con-trols contained DMSO as vehicle only. For the isobolograms, three fixed combination ratios (enzalutamide:pimozide: 1:1, 1:2, 1:5) and enzalutamide concentrations ranging from 0.01 to 100 μM were chosen.

## Western blot analysis

For western blot analysis cells were washed with phosphate-buffered saline (PBS), harvested by using a cell lifter, and lysed in Radioimmunoprecipitation assay (RIPA) buffer with complete Mini EDTA-free protease inhibitor tablets (Roche) and phosphatase inhibitor cocktail Phos-STOP (Roche). The protein concentration was quantified using the BCA Assay (Thermo-Fischer Scientific) as described earlier [37]. 20 μg protein lysate were separated by SDS-gel electrophoresis using a NuPAGE™ 4–12% Bis-Tris protein gel and transferred to a nitrocellu-lose membrane using the iBlot Dry Blotting System (all ThermoFischer Scientific). As protein standards, 10 μl Spectra Multicolour Broad Range (ThermoFisher Scientific) and 1 μl Magic-Mark™ XP Western Protein Standard (ThermoFisher Scientific) were used. For detection, the membranes were incubated with WesternBright Sirius HRP substrate (Advansta). Except for the membranes displayed in S2C Fig, all signals were detected by a Microchemi chemilumines-cence system (DNR Bio-Imaging Systems). S2C Fig had been digitalised by using an Odyssey CT (LI-COR). Antibodies used are listed in S2 Table. Densitometric analysis of experiments

was made with the Image-Studio Lite 5.2 software (LI-COR). Uncropped western blot images are displayed in the supplementary files. Raw images files are displayed in S1 Raw images.

## Subcellular fractionation

For subcellular fractionation, cells were washed with ice-cold PBS and directly harvested in 400 μl cytoplasmic lysis buffer (10 mM HEPES pH 7.9, 10 mM KCl, 1.5 mM $MgCl_2$, 340 mM Sucrose, 10% Glycerol, 1 mM DTT, protease inhibitor, phosphatase inhibitor) before the addition of 0.1% Triton X-100 and incubation for 5 min on ice. 200 μl of the suspension was used for whole-cell lysates. To isolate the nuclear fraction, the remaining 200 μl were collected by centrifugation at 1300 x g for 4 min at 4˚C and separated from the cytoplasmic proteins in the supernatant. After separation, sample buffer was added and the samples were sonicated. 20 μl of the fractions were used for western blot analysis. Fraction quality were controlled by detection of Lamin A/C (nuclear fraction) and GAPDH (cytoplasmic fraction).

## RNA isolation and quantitative real-time PCR

Cells were seeded at a density of 500,000 cells/well in 6 well plates and treated after 24 h. Cells were harvested 48 h after treatment. Total RNA was isolated by using the RNeasy Plus Mini Kit following the manufacturer's instructions (Qiagen). cDNA synthesis was performed using the iScript Select cDNA synthesis kit (Bio-Rad). qPCR was performed using the MIC qPCR cycler (BioMolecular Systems) and TaqMan gene expression assays for STAT5a (Hs00559637_g1), STAT5b (Hs00560026_m1), PSA/KLK3 (Hs02576345_m1), Bcl-xL (Hs00236329_m1), Cyclin D1 (Hs01050839_m1), and HPRT1 (Hs02800695_m1, all Applied Biosystems). HPRT1 was used as a reference. The micPCR software was used for the determination of Ct values. $\Delta Ct = Ct_{GOI} - Ct_{HPRT1}$ values were calculated and expressed as $2^{-\Delta Ct}$.

## siRNA transfections

For siRNA transfections, 25 nM ON-TARGETplus SMARTpool against STAT5a or STAT5b (both Dharmacon) were used. As a control, the ON-TARGETplus Non-targeting Control Pool was used (Dharmacon). Reverse transfections of siRNA were performed with Lipofectamine RNAiMAX (Invitrogen) according to the manufacturer's protocol.

## Measurement of cell viability

Cell viability was assessed using the 3-(4,5-dimethylthiazol- 2-yl)-2,5-diphenyltetrazolium bromide (MTT) dye reduction assay (Roche). Cells (5000 cells in 50 μl) were seeded into 96-well culture plates. After 24 h, the cells were treated by adding the drug at different concentrations in 50 μl of the medium. After 72 h treatment, MTT (0.5 mg/ml) was added for additional 4 h. After that, the cells were lysed in a solution containing 10% SDS in 0.01 M HCl. The plates were incubated overnight at 37˚C, 5% $CO_2$. Absorbance was recorded at 570 nm for each well using a SPARK 10M Microplate Reader (Tecan). Each experiment was done in triplicate. After subtracting background absorbance, results were calculated as x-fold of untreated control cells.

## Statistical analysis

Prism 8.3 (GraphPad Software) and SPSS Statistics 25 (IBM) was used for statistical analyses. For curve fitting of dose-response curves, non-linear regression was used. Differences between treatment groups were analysed using Student's t-test. Data are presented as mean±s.e.m. to estimate the various means in multiple repeated experiments [38]. P-values of ≤0.05 were

considered to be statistically significant. All differences highlighted by asterisks were statistically significant as encoded in figure legends (*p≤0.05; **p≤0.01; ***p≤0.001). All experiments have been performed in at least three biological replicates unless noted otherwise. Combination Index (CI) was calculated using the following equation: $CI = (C_{A,X}/IC_{X,A})+(C_{B,X}/IC_{X,B})$ [39].

## Results

### STAT5 is highly heterogeneously expressed in enzalutamide-resistant cell models

To assess STAT5a and b expression in enzalutamide-resistant cell models, publicly available data sets were analysed for STAT5a and STAT5b mRNA expression. mRNA expression analysis of enzalutamide-sensitive and enzalutamide-resistant CRPC xenografts derived from LNCaP (GSE55345) cells revealed that STAT5a had a mean relative mRNA expression of 0.05279 with a variance of 0.000 in enzalutamide-sensitive CRPC samples and a mean of 0.06002 with a variance of 0.003 (Fig 1A, S1E Fig) [40]. mRNA analysis of the same dataset for

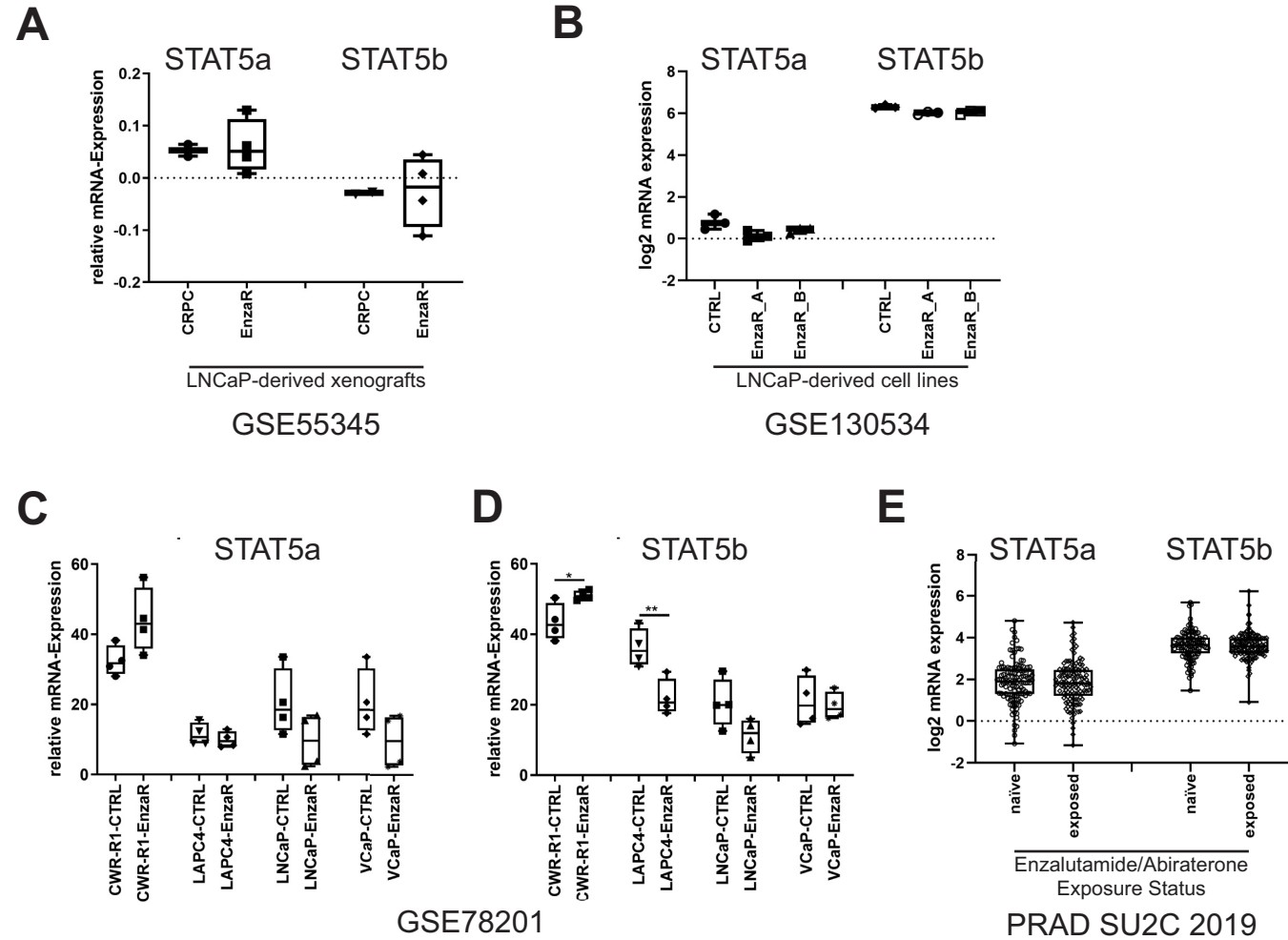

**Fig 1. STAT5 expression is heterogeneous in enzalutamide-resistant cell models.** Analysis of STAT5a and STAT5b mRNA expression in several datasets of (A) LNCaP-derived xenografts (GSE55345), (B) LNCaP-derived cell lines (GSE130534), (C+D) enzalutamide-resistant AR-positive PCa cell lines (GSE78201), and (E) abiraterone/enzalutamide therapy naïve (n = 121) and exposed (n = 128) mCRPC patients (PRAD SU2C 2019).

STAT5b revealed a mean of -0.02889 with a variance of 0.000 in enzalutamide-sensitive CRPC samples and a mean of -0.02889 with a variance of 0.005 (Fig 1A, S1E Fig). In LNCaP-derived cell line models created by enzalutamide dose escalation (GSE130534), STAT5a and STAT5b mRNA expression were not altered (Fig 1B) [41]. Analysis of different enzalutamide-resistant cell line models (GSE78201) also demonstrated a heterogeneous expression of STAT5a (Fig 1C) and STAT5b (Fig 1D). Moreover, analysis of the dataset (GSE78201) also revealed that regulation of STAT5a and STAT5b is not uniform between parental and enzalutamide-resistant cells [17] (Fig 1C and 1D). Compared to their enzalutamide-sensitive controls, the enzalutamide-resistant CWR-R1 cell line showed an increase in STAT5a/b mRNA levels, whereas enzalutamide-resistant LAPC4 and LNCaP cells showed a decrease in STAT5a/b expression. Enzalutamide-resistant VCaP cells displayed a reduction in STAT5a mRNA expression compared to their enzalutamide-sensitive control cell line (Fig 1C), whereas STAT5b mRNA levels did not alter between the cell lines (Fig 1D). Analysis of the dataset published by Abida et al. revealed a highly heterogeneous expression of STAT5 in patients with mCRPC, who were naïve or exposed to hormone therapy (Fig 1E) [42]. However, there were no statistically significant differences between STAT5a or STAT5b mRNA expression between the groups.

For the experiments in this study, the LNCaP sub-lines C4-2 (enzalutamide-sensitive) and MR49F (enzalutamide-resistant) and the LAPC4 sub-lines LAPC4-CTRL (enzalutamide-sensitive) and LAPC4-EnzaR (enzalutamide-resistant) were chosen.

STAT5a/b protein levels were markedly higher in MR49F compared to C4-2 cells (Fig 2A and 2B), whereas the STAT5a/b protein levels in enzalutamide-resistant LAPC4-EnzaR were similar to those in enzalutamide-sensitive LAPC4-CTRL. Both tested enzalutamide-resistant cell lines showed an increase in nuclear STAT5 level (Fig 2C and 2D, S2B Fig). Since the used antibody can not distinguish between both forms of STAT5, we additionally evaluated the mRNA expression of the STAT5a and STAT5b genes. STAT5a mRNA levels were significantly lower in MR49F cells compared to C4-2 cells (Fig 2E). In contrast, STAT5b mRNA levels were significantly elevated in MR49F cells (Fig 2F). The same trend was indicated by the NGS data published by King et al., who compared MR49F cells to the enzalutamide-sensitive cell line V16D (S1F Fig) [7]. The LAPC4 sub-lines expressed STAT5a mRNA at a low level and a very high level of STAT5b mRNA (Fig 2E and 2F). When comparing AR and PSA/KLK3 between C4-2 and MR49F cells, the enzalutamide-resistant MR49F cells showed decreased AR protein and PSA/KLK3 protein levels (Fig 2G and 2H). Compared to the LAPC4-CTRL cells, the LAPC4-EnzaR cell line displayed a decrease in AR protein level and no PSA/KLK3 protein expression at all (Fig 2G and 2H). Another member of the STAT protein family, STAT3, has also been implicated in enzalutamide resistance [43]. Comparison of STAT3 expression in enzalutamide-resistant cell models to its enzalutamide-sensitive controls revealed that STAT3 was not increased in the enzalutamide-resistant models (S2A and S2B Fig). Constitutively active STAT3 measured by phosphorylation of its tyrosine 705 was also not detectable. To test if STA3 can be activated, the LAPC4- and DuCaP-derived cell lines were treated with IL6, a known cytokine to activate STAT3 [44]. In LAPC4-derived models, IL6 did not induce treatment-induced STAT3 phosphorylation at all, whereas the LNCaP- and DuCaP-derived models showed a distinct pSTAT3 signal after IL6 treatment (S2C Fig). Taken together, STAT5 does not follow any expression pattern in enzalutamide-resistant PCa cell lines.

## The androgen receptor does not directly regulate STAT5a or STAT5b

Next, we assessed whether the altered expression of STAT5a or STAT5b in MR49F was a direct transcriptional effect of the AR. To this end, publicly available AR chromatin immunoprecipitation sequencing (ChIP-Seq) datasets (GSE62442, GSE65066, Fig 3) were analysed for the loci

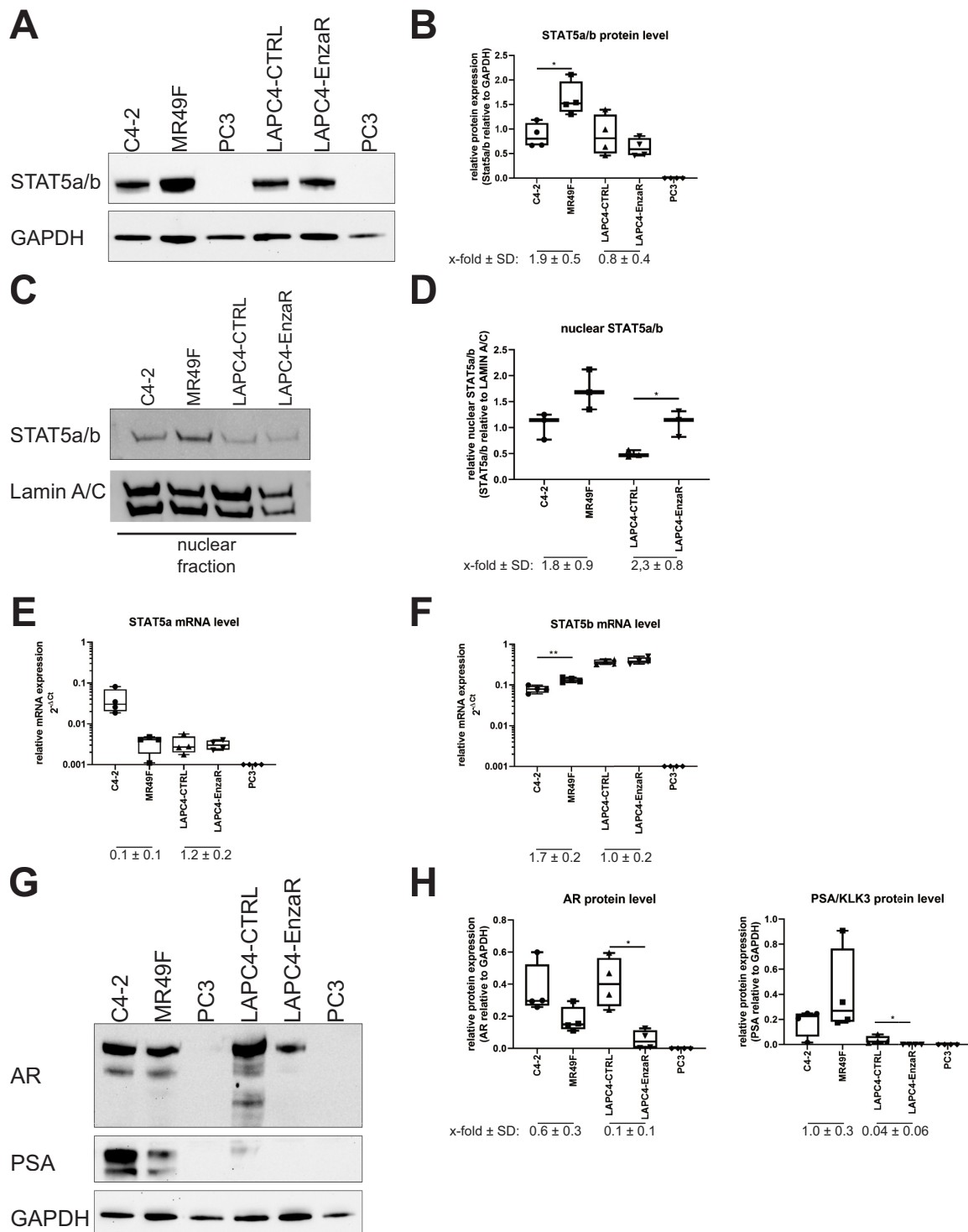

**Fig 2. STAT5 expression in enzalutamide-resistant cell models derived from LNCaP and LAPC4.** (A) Representative western blot of STAT5a/b and GAPDH expression in C4-2, MR49F, LAPC4-CTRL, LAPC4-EnzaR and PC3 cells. PC3 cells were used as a negative control for STAT5 and AR expression. (B) Densitometric analysis of STAT5a/b protein expression levels relative to GAPDH. Data is shown as box plot with whiskers (min to max) of at least three independent western blot experiments. (C) Representative western blot of nuclear STAT5a/b and Lamin A/C protein in C4-2, MR49F, LAPC4-CTRL, LAPC4-EnzaR cells. (D) Densitometric analysis of STAT5a/b protein expression levels relative to Lamin A/C. Data is

shown as box plot with whiskers (min to max) of at least three independent western blot experiments. (E) STAT5a and (F) STAT5b mRNA levels were measured by qPCR and normalised to HPRT1. Relative expression levels in C4-2, MR49F, LAPC4-CTRL, LAPC4-EnzaR and PC3 cells (STAT5 negative control) are shown as mean±s.e.m. of at least three independent experiments. (G) Representative western blot of AR, PSA/KLK3 and GAPDH protein expression in the cell lines C4-2, MR49F, LAPC4-CTRL, LAPC4-EnzaR and PC3 (negative control for AR, PSA/KLK3, and STAT5). (H) Densitometric analysis of AR and PSA/KLK3 protein expression levels relative to GAPDH. Data is shown as box plot with whiskers (min to max) of at least three independent western blot experiments. (*: p ≤0.05 **:p≤0.01 ***: p ≤0.001). Uncropped western blots are displayed in S1G, S1H and S2A Figs.

of STAT5a, STAT5b, and PSA/KLK3 (positive control) [31]. In the genomic region surrounding STAT5a/b, we found three AR binding sites (ARBs) which were not detected in any of examined cell lines (Fig 3A). In C4-2 cells, these ARBs showed weak AR binding induced by the synthetic androgen Mibolerone (Fig 3B). In comparison to this, the ARBs in the vicinity of the well-known AR target gene *PSA/KLK3*were found in all shown cell lines (Fig 3C) and showed much stronger Mibolerone-induced AR binding in C4-2 cells as it was seen at the genomic region surrounding STAT5a/b (Fig 3D). To validate these findings at mRNA level, AR signalling was modulated in C4-2 cells by steroid-starvation for 24 h followed by a 48 h treatment with 1 nM of the synthetic androgen R1881, or 1 nM R1881 plus 1 μM and 10 μM enzalutamide. None of the treatments led to a significant change in STAT5a or STAT5b mRNA levels. In contrast, the control (PSA/KLK) showed a significant increase with R1881 treatment which could be reverted by enzalutamide (Fig 3E). To verify the findings in C4-2 cells, several publicly available datasets (GSE40050, GSE81796, GSE69896, GSE78201) were analysed for STAT5a and STAT5b after enzalutamide treatment (Fig 3F). None of the analysed datasets showed a significant change in STAT5a and STAT5b mRNA levels after enzalutamide treatment. STAT5a and STAT5b mRNA levels in LNCaP and the CRPC LNCaP sub-lines LNCaPabl and C4-2 were assessed in several datasets (GSE40050, GSE11428, GSE13332) after siRNA-mediated AR–knockdown. This analysis also revealed no significant change in STAT5b mRNA levels after AR knockdown (Fig 3G). Similarly, STAT5a levels showed no significant increase in LNCaP and LNCaPabl cells after AR-knock-down. In contrast, C4-2 displayed a significant decrease in STAT5a levels after AR-knock-down. Taken together, these results indicate that androgens do not regulate STAT5a and STAT5b by a direct transcriptional mechanism.

## Isobologram analysis of combined STAT5 inhibitor pimozide and enzalutamide treatment

Combination drug regimens for the treatment of cancer often achieve a therapeutic efficacy greater than that observed with monotherapy. In the search for more effective treatments, we tested the FDA-approved STAT5 inhibitor pimozide in combination with enzalutamide. Western blot analysis demonstrated a decrease in total STAT5 levels in MR49F and C4-2 cells (S3A Fig) whereas overall STAT5 levels in the LAPC4 models were not changed after pimozide treatment (S4A Fig). In addition, a concentration-dependent decrease of nuclear STAT5 after pimozide treatment could be detected in all tested cell lines (S3B, S3C, S4B and S4C Figs). Moreover, we tested the inhibitory effects on STAT5 at 10 μM pimozide by measuring the mRNA expression of its target genes Cyclin D1 (*CCND1*) and BCL-xL (*BCL2L1*) after treatment. The qPCR analysis showed a decrease of at least one of the target genes in C4-2 and MR49F cells after pimozide treatment thus indicating a STAT5 inhibition (S5B Fig).

The mean $IC_{50}$ values of cell viability for C4-2, MR49F, LAPC4-CTRL, and LAPC4-EnzaR cells (Fig 4A–4D) were examined for their sensitivities to enzalutamide and pimozide and are presented in Table 1. The dose-response curves revealed that with a higher ratio (Enza+Pimo), cell viability decreased and cells became more sensitive to lower enzalutamide concentrations

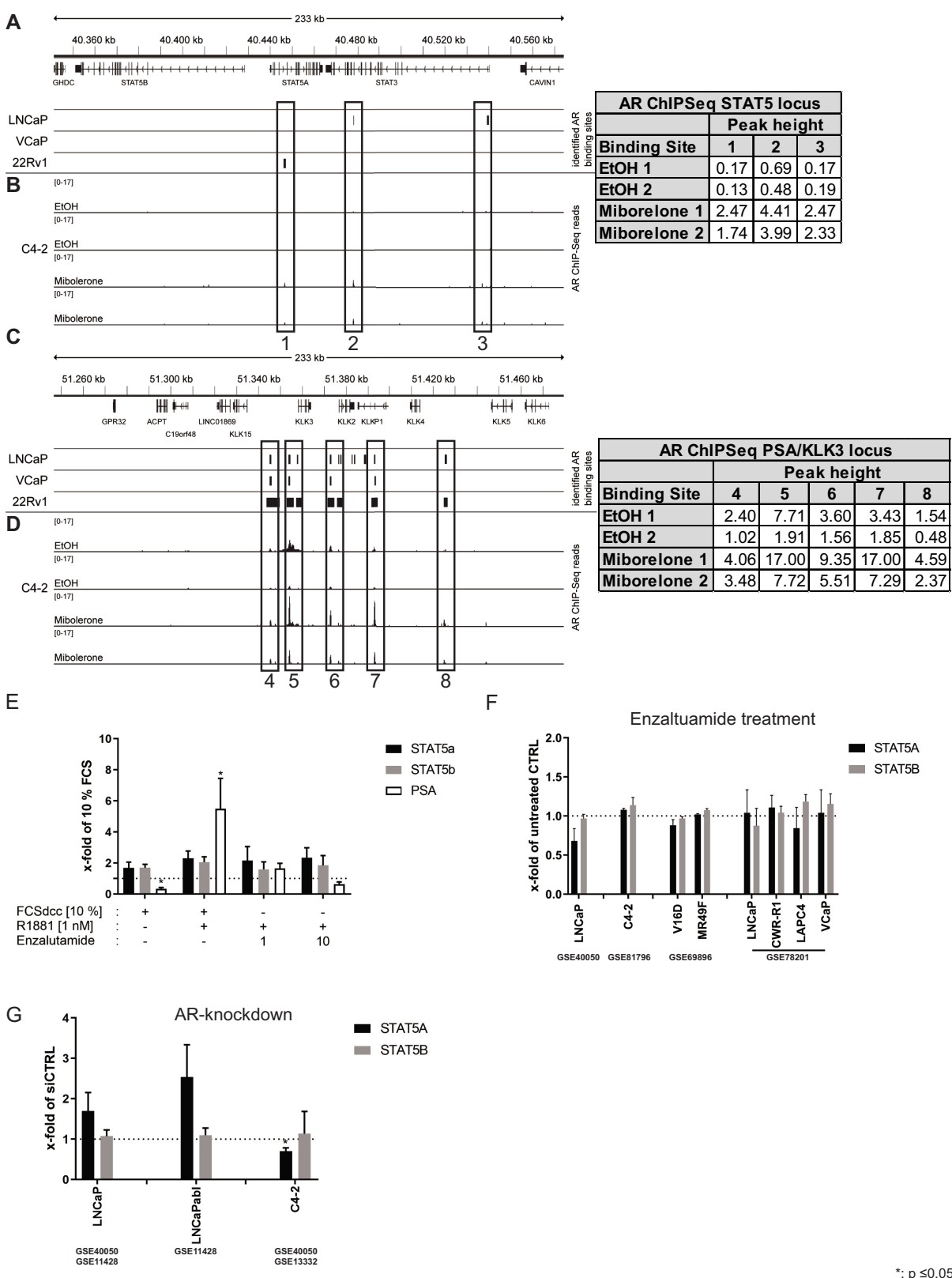

**Fig 3. AR does not directly regulate STAT5 expression.** (A) Visualisation of the AR binding sites from a publicly available ChIP-Seq dataset (GSE62442) at the STAT5a and STAT5b locus of the AR-positive cell lines LNCaP, VCaP, and 22Rv1 which were treated with vehicle or 1 nM R1881 for 1 h (B) Visualisation of AR chromatin binding and normalised peak heights from a publicly available ChIP-Seq dataset (GSE65066) at the STAT5a and STAT5b locus of the cell line C4-2, which was treated with vehicle (EtOH) or 10 nM of the synthetic androgen Mibolerone. (C) Analysis of ChIP-Seq dataset (GSE62442) at the *PSA/KLK3* locus of the AR-positive cell lines LNCaP, VCaP, and 22Rv1 that were treated with vehicle or 1 nM R1881 for 1 h (D) Analysis of ChIP-Seq dataset (GSE65066) at the *PSA/KLK3* locus of the cell line C4-2, which was treated with vehicle or 10 nM of the synthetic androgen Mibolerone. (E) Validation of the AR-binding sites at the *PSA/KLK3* and *STAT5* loci achieved from the ChIP-Seq datasets in C4-2 cells on transcript level. Cells were cultured in 10% charcoal-stripped FCS for 24 h followed by a 48 h treatment with 1 nM R1881, 10% FCS, or with 1 nM R1881 combined with 1 μM and 10 μM enzalutamide. (F+G) Analysis of STAT5a and STAT5b mRNA expression in several datasets of (F) enzalutamide-sensitive and enzalutamide-resistant cell lines after enzalutamide treatment (GSE40050, GSE81796, GSE69896, GSE78201) and (G) enzalutamide-sensitive PCa cell lines after siRNA-mediated AR-knock-down (GSE13332, GSE11428, GSE40050).

(Fig 4). This result is also reflected by the decrease of the IC$_{50}$ values with increased pimozide concentration (Table 1). The CI of the LNCaP sub-lines C4-2 and MR49F demonstrated clearly that all treatments were synergistic on cell viability (CI<1; Fig 4A and 4B). In contrast, the LAPC4 sub-lines LAPC4-CTRL (Fig 4C) and LAPC4-EnzaR (Fig 4D) showed only additive effects of enzalutamide and pimozide on cell viability.

## Influence of STAT5a- and STAT5b-knockdown on cell viability in the presence and absence of enzalutamide

Several studies have demonstrated that next to the ability of pimozide to inhibit STAT5a/b phosphorylation and function it can also exhibit little activity against other kinases and transcription factors such as STAT1 [45, 46]. Therefore, the results obtained with pimozide were validated by a specific siRNA-mediated knockdown of STAT5a and STAT5b (S6 Fig). The qPCR analysis showed that both siRNA treatments specifically reduced the mRNA expression of their targets (S6A and S6B Fig). The STAT5a-directed siRNA showed only a marginal effect on STAT5b mRNA levels (S6A Fig). The qPCR data also revealed that the siRNA-mediated knockdown was already achieved 24 h after transfection and lasted at least over 72 h (S6C and S6D Fig). Western Blot analysis also revealed an effective knockdown of STAT5b protein level in C4-2, MR49F, LAPC4-CTRL and LAPC4-EnzaR cells (S6E and S6F Fig). To investigate the role of STAT5a/b in cell viability, MTT assays have been performed with the established siRNA (Fig 5). In the absence of enzalutamide, both STAT5a- and STAT5b-directed siRNA pools induced only negligible effects on cell viability compared to a control siRNA pool (siCTRL) after 72 h (Fig 5A and 5B). Enzalutamide treatment of C4-2 cells transfected with siCTRL reduced cell viability significantly by 30% (Fig 5A). Knock-down of STAT5a or STAT5b in enzalutamide treated C4-2 cells showed no additional effect on the enzalutamide-induced decrease in cell viability. In contrast, enzalutamide had a minor impact in MR49F cells transfected with siCTRL or the siSTAT5a pool (Fig 5B). However, enzalutamide-treated MR49F cells transfected with siRNAs against STAT5b showed a decrease in cell viability to approximately 70% of the control (untreated MR49F cells transfected with siCTRL). Based on these observations, dose-response experiments with concentrations from 0.001 to 10 μM enzalutamide with knockdown of STAT5a or STAT5b cells were performed in MR49F (Fig 5C). As reported in Fig 5B, MR49F cells transfected with siCTRL or the siSTAT5a pool showed negligible effects on viability after enzalutamide treatment. In MR49F cells transfected with the siSTAT5b pool, a significant decrease in cell viability in response to enzalutamide (concentrations of 0.1 μM and higher) was observed. To test if STAT5-knockdown is able to resensitise the enzalutamide-resistant model LAPC4-EnzaR, the cells and their enzalutamide-sensitive control were transfected with siCTRL, siSTAT5a or siSTAT5b and treated with vehicle or 10 μM enzalutamide (Fig 5D and 5E). Both LAPC4 sublines showed a decrease in cell viability after transfection with siSTAT5a in presence and absence of enzalutamide. In line with the

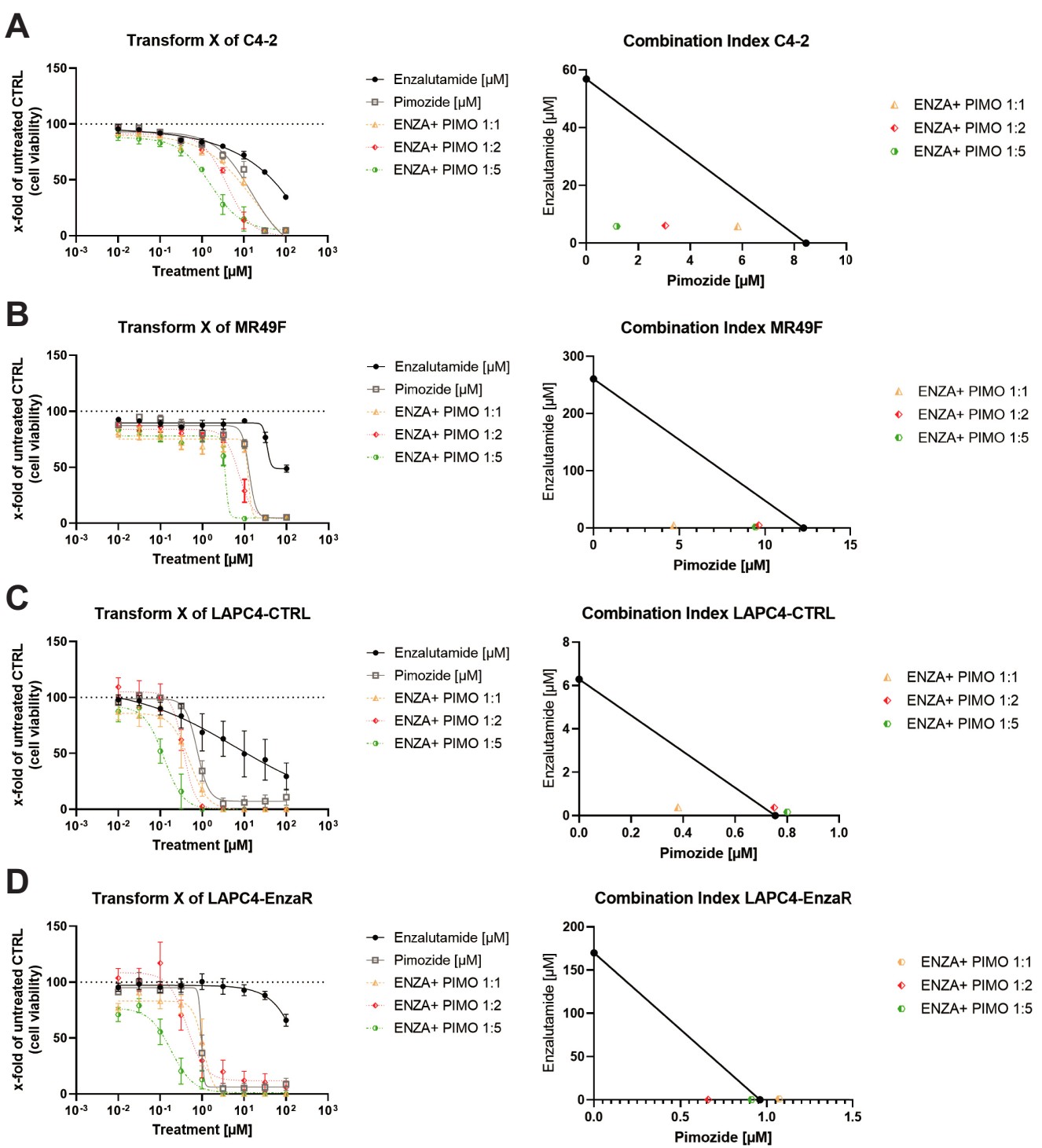

**Fig 4. Isobologram analysis of combined treatment with the STAT5 inhibitor pimozide and enzalutamide.** Isobologram experiments were carried out with enzalutamide and pimozide as introduced by Loewe [36]. Results of dose-response experiments and graphical illustration of the combination index (CI) in C4-2 cells (A), MR49F cells (B), LAPC4-CTRL (C), and LAPC4-EnzaR cells (D). Cell viability after enzalutamide and pimozide treatment was assessed by MTT viability assays. Data is shown as mean±s.e.m. of three independent experiments.

**Table 1. Summary of isobologram analysis.**

| C4-2 | IC$_{50}$ | COMBINATION | | | |
|---|---|---|---|---|---|
| Enzalutamide (μM) | 56.83 | Ratio | Enzalutamide (μM) | Pimozide (μM) | CI |
| Pimozide (μM) | 8.451 | 1:1 | 5.83 | 5.83 | 0.79 |
| | | 1:2 | 3.04 | 6.08 | 0.77 |
| | | 1:5 | 1.17 | 5.84 | 0.71 |
| MR49F | IC$_{50}$ | COMBINATION | | | |
| Enzalutamide (μM) | 260.7 | Ratio | Enzalutamide (μM) | Pimozide (μM) | CI |
| Pimozide (μM) | 12.24 | 1:1 | 4.67 | 4.67 | 0.40 |
| | | 1:2 | 4.81 | 9.63 | 0.81 |
| | | 1:5 | 1.89 | 9.44 | 0.78 |
| LAPC4-CTRL | IC$_{50}$ | COMBINATION | | | |
| Enzalutamide (μM) | 11.11 | Ratio | Enzalutamide (μM) | Pimozide (μM) | CI |
| Pimozide (μM) | 0.713 | 1:1 | 0.38 | 0.38 | 0.56 |
| | | 1:2 | 0.37 | 0.75 | 1.08 |
| | | 1:5 | 0.16 | 0.80 | 1.13 |
| LAPC4-EnzaR | IC$_{50}$ | COMBINATION | | | |
| Enzalutamide (μM) | 169.9 | Ratio | Enzalutamide (μM) | Pimozide (μM) | CI |
| Pimozide (μM) | 0.911 | 1:1 | 0.73 | 0.73 | 0.80 |
| | | 1:2 | 0.35 | 0.69 | 0.76 |
| | | 1:5 | 0.20 | 1.01 | 0.08 |

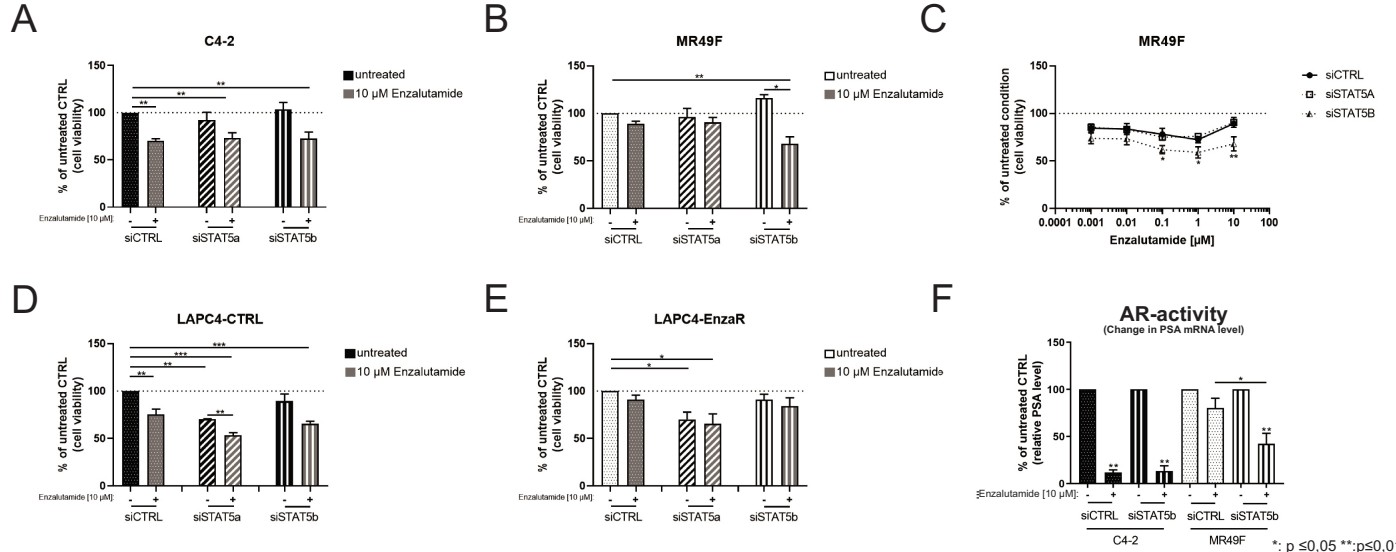

**Fig 5. Influence of STAT5a and STAT5b knockdown on cell viability in the presence and absence of enzalutamide.** (A+B) Cell viability of C4-2 cells (A) and MR49F cells (B) after knockdown of STAT5a or STAT5b in the presence and absence of 10 μM enzalutamide was assessed by MTT assays. Data were plotted as mean±s.e.m. of three independent experiments. (C) Dose-response curve of MR49F cells with knocked-down STAT5a and STAT5b treated with enzalutamide (0,001 to 10 μM). Cell viability was assessed by MTT assays. Data is shown as mean±s.e.m. of three independent experiments. (C+D) Cell viability of LAPC4-CTRL cells (D) and LAPC4-EnzaR cells (E) after knockdown of STAT5a or STAT5b in the presence and absence of 10 μM enzalutamide was assessed by MTT assays. Data is plotted as mean±s.e.m. of three independent experiments. (*: p ≤0.05 **:p≤0.01 ***: p ≤0.001) (F) PSA/KLK3 regulation as an indicator for AR activity in C4-2 cells and MR49F cells after siCTRL and siSTAT5b pool transfection in the presence and absence of 10 μM enzalutamide was measured by qPCR. Data is shown as mean as mean±s.e.m. of three independent experiments (*: p ≤0.05 **:p≤0.01 ***: p ≤0.001).

results in C4-2 cells, cell viability in LAPC4-CTRL was significantly reduced (~ 70% of untreated control) and siSTAT5b did not show any further reduction (Fig 5D). However, in contrast to the MR49F cell line, siRNA-mediated STAT5b-knockdown was not able to resensitise LAPC4-EnzaR cells to the enzalutamide treatment (Fig 5E). Taken together, the LNCaP-derived cells C4-2 and MR49F were not influenced by siRNA knock-down alone and the enzalutamide-resistant MR49F could be sensetised to enzalutamide by STAT5b knock-down. In contrast, the LAPC4 sub cell lines showed a reduced cell viability after STAT5a knock-down and the combination of enzalutamide with STAT5a or STAT5b knock-down has no further effects on cell viability.

## Enzalutamide reduced AR activity after knockdown of STAT5b in MR49F cells

To examine whether the effects on cell viability after enzalutamide treatment and STAT5b knockdown were due to regulation of AR activity, we tested AR activity indirectly by measuring mRNA expression of the clinically relevant AR-target gene PSA/KLK3by qPCR. In the absence of enzalutamide, STAT5 knockdown had no significant effect on the PSA/KLK3 mRNA expression compared to the control in the LNCaP sub-lines (S5B Fig). In the presence of 10 μM enzalutamide, C4-2 cells transfected with siCTRL or the siSTAT5b pool showed reduced PSA/KLK3 expression to levels less than 10% of those measured in controls (Fig 5F). In contrast, MR49F cells transfected with siCTRL showed only a minor reduction of PSA after enzalutamide treatment. However, the PSA/KLK3 level was reduced to ~36% after enzalutamide treatment in MR49F cells transfected with the siSTAT5b pool (Fig 5F). Taken together, AR activity and cell viability can be re-established to enzalutamide by STAT5b knock-down in MR49F cells.

## Discussion

The non-steroidal second-generation anti-androgen enzalutamide is a promising agent for patients with pre- and post-chemotherapy CRPC and demonstrated to improve survival; however, relapse and development of therapy resistance remain the major reasons for poor survival [4, 5, 47–49]. Managing enzalutamide-resistance is still a challenge. In a recent study by Udhane et al., the JAK2/STAT5 pathway could be identified as a possible target during enzalutamide-resistant development [50]. In line with the study by Thomas et al., the group showed an increase in STAT5 protein and STAT5 activity expression during and after hormone therapy [28, 50].

The mRNA expression analysis presented here showed that STAT5a and STAT5b levels are heterogeneous in different enzalutamide-resistant cell lines and xenograft models. Moreover, mRNA expression analysis of hormone-naïve and treated mCRPC patients also showed a heterogeneous mRNA expression profile of STAT5a or STAT5b [16]. However, no differences could be demonstrated in STAT5a or STAT5b mRNA expression between the two patient groups. As the AR is targeted by enzalutamide directly, ChiP-Seq database analysis at the STAT5 locus has been performed. These results revealed there was no AR binding site around this locus. To underpin the ChIP-Seq results, we analysed AR-positive PCa cell lines after chemical AR-blockade with enzalutamide or AR-knock-down with siRNA. The results revealed that there was none or no uniform regulation of the STAT5a/b mRNA.

As shown here STAT5a or STAT5b are not directly regulated by the AR (Fig 3). The discrepancy between the protein and mRNA levels may be explained by the influence of the phosphorylated tyrosine 694 on STAT5a/b protein stability [51]. This view is supported by the data from Udhane et al. who revealed an upregulation and activation of JAK2 by AR in the presence

of enzalutamide, thus leading to an increase in STAT5a/b phosphorylation [50, 52, 53]. Our protein analysis of the enzalutamide-resistant MR49F and LAPC4-EnzaR revealed an increase of nuclear STAT5a/b protein compared to their enzalutamide-sensitive control cell lines. Nuclear localisation has been shown to reflect STAT5a/b activity as STAT5a/b phosphorylation is mandatory for STAT5 to translocate into the nucleus [50, 52]. STAT3 has been shown to be involved in enzalutamide-resistance. Constitutively active STAT3 has been reported to be involved in enzalutamide-resistance by inducing a neuroendocrine differentiation [43, 54]. Neither the analysis of pSTAT3 shown here nor the study from Udhane et al. could reveal constitutively activated STAT3 in the enzalutamide-resistant models [50]. Luo et al. have demonstrated that modulation of the STAT3 pathway leads to neuroendocrine differentiation. As the models used here are all AR-positive, neuroendocrine differentiation can be excluded [54, 55]. Therefore, phosphorylated STAT3 does not seem to be involved in enzalutamide-resistance. These results demonstrate again the complex mechanisms leading to enzalutamide-resistance in PCa [19].

The STAT5 inhibitor pimozide is an FDA-approved compound used to clinically treat chronic psychosis, Tourette syndrome, and resistant tics [56]. Moreover, in previous studies pimozide was shown to have an anti-cancer effect on various tumour entities, including PCa [45]. The isobolograms of combination treatments showed that increasing concentrations of pimozide with enzalutamide were leading to a synergistic inhibitory effect on cell viability in LNCaP sub-lines. In contrast to the findings in C4-2 and MR49Fcells, the LAPC4 sub-lines showed an additive effect on cell viability after treatment with enzalutamide and pimozide. The activity of pimozide against other kinases and transcription factors such as NFκB, STAT1 or STAT3 and the different molecular background of the cells may explain this difference [45, 46]. All tested cell lines showed a decrease in cell viability when treated with enzalutamide and pimozide. This result is in line with the data shown by Udhane et al. who demonstrated reduced tumour growth by cell death after treatment with the combination of STAT5 inhibitors and enzalutamide compared to single treatment [50].

Due to activity of pimozide against other kinases and to gain further insight into the molecular mechanisms of the enzalutamide/pimozide treatment, the impact of STAT5a/b knockdown in combination with enzalutamide on cell viability was assessed. In the absence of enzalutamide the knock-down of neither STAT5a nor of STAT5b showed effects on the cell viability of C4-2 and MR49F cells whereas the LAPC4-derived cells show a decrease after STAT5a knockdown. This result is contradictory to the published results from Ahonen et al. who postulated that STAT5a/b inhibition causes apoptosis in PCa cells [10]. It has to be kept in mind that Ahonen and colleagues used different cell lines and a dominant-negative overexpression plasmid for inhibition of STAT5a/b. Therefore, due to the variances in experimental set up different results are possible. Furthermore, in the presence of 10 μM enzalutamide, the STAT5a or STAT5b knock-down had no additional effect in C4-2. These results lead to the conclusion that the synergistic effects seen in the isobolograms in C4-2 cells with the enzalutamide and pimozide combination were due to off-target effects of pimozide, as they have been reported previously [23, 24]. However, in MR49F cells STAT5b knock-down and treatment with 10 μM enzalutamide lead to a reduction of cell viability to 70%. These inhibitory effects on cell growth were comparable to enzalutamide-sensitive C4-2 cells. The same result was shown in a dose-response curve which indicates that MR49F cells with knocked-down STAT5b are already showing a dose-dependent reduction of cell viability due to enzalutamide.

Several studies demonstrated that the STAT5 pathway is not only functionally synergistic with the AR pathway but is also involved in the regulation of the cell cycle and apoptosis by increasing the expression levels of cyclin D1 and Bcl-xL [57–59]. Therefore, STAT5 may play a key role in PCa progression. Complete STAT5-knock-down could cause a reduction in cell

viability [28, 50, 60]. Here, in the absence of enzalutamide, the siRNA-mediated knockdown of STAT5b had negligible influence on cell viability in the tested cell lines. Knock-down of STAT5a did not have any impact on enzalutamide-resistant MR49F cells but lead to a decrease in cell viability in both LAPC4 sub-lines. In CRPC cell lines, STAT5a has been shown to be a potential therapeutic target, and its inhibition induced apoptosis probably by regulation of anti-apoptotic proteins [25, 61]. Besides, STAT5a has also been reported to interact with the GR [62, 63]. Puhr et al. already reported an essential role of the GR in the LAPC4 sub-lines [21]. Therefore, the effects seen in these LAPC4 sub-lines after STAT5a knockdown may occur due to its influence on the GR [21, 62].

Our investigation on cell viability was extended by assessing AR activity after STAT5b knockdown and enzalutamide treatment. As expected, C4-2 cells showed a decrease in PSA/ KLK3 to expression levels below 10% compared with those measured in untreated cells. In contrast, PSA/KLK3 reduction was only observed in MR49F cells when STAT5b was knocked down. This result is in line with those presented by Mohanty et al. showing no relationship between STAT5a expression and AR activity [61]. The results on cell viability in combination with the reduction in AR activity lead to the conclusion that siRNA-mediated STAT5b-knock-down can resensitise the MR49F cell line to enzalutamide treatment.

Tumour heterogeneity is one of the biggest challenges in cancer therapy [64]. The different molecular adaptations are one of the major problems in the development of new therapies [8, 9, 64]. Several groups have suggested STAT5 as a target in late-stage PCa [28, 50, 52, 61, 65]. The results presented here show that STAT5a and STAT5b mRNA expression is highly hetero-geneous in enzalutamide-resistant PCa cell lines, whereas STAT5a/b activity seems to be increased in the tested cell lines. Effects of specific STAT5a/b knockdown on cell viability and AR signalling also seem to be dependent on the cellular background of the resistant cells, reflecting the challenges of tumour heterogeneity in therapy treatment of late-stage diseases again.

Interestingly, chemical inhibition of STAT5 activity by pimozide combined with enzaluta-mide showed synergistic effects on the reduction of cell growth. However, it has to be men-tioned, that these synergistic effects of pimozide may result from off-target effects of the inhibitor as it has been described before [45, 46, 61]. Also, more studies in *in vivo* and *ex vivo* models are necessary to clarify the role of STAT5 in enzalutamide resistance. In conclusion, based on the *in vitro* data presented herein its role in enzalutamide-resistance and its potential as a therapeutic target could not be tightened.

## Supporting information

**S1 Fig. Graphical description of the enzalutamide-resistant cell models, dose-response curves and STAT5 analysis.** (A) Graphical description of the establishing process of C4-2 and MR49F cells. (B) Results of enzalutamide dose-response experiments in C4-2 and MR49F cells on cell viability 72 h after treatment. Cell viability was assessed by MTT assays. Data is shown as mean±s.e.m. of three independent experiments. (C) Graphical description of the establish-ing process of LAPC4-CTRL and LAPC4-EnzaR cells. (D) Results of enzalutamide dose-response experiments in LAPC4-CTRL and LAPC4-EnzaR cells on cell viability 72 h after treatment. Cell viability was assessed by MTT assays. Data is shown as mean±s.e.m. of three independent experiments. (E+F) STAT5a and STAT5b expression analysis of the public data set of the LNCaP-derived xenografts (E, GSE55345) and of the CRPC cell models V16D and MR49F (F, GSE87153) [7, 15, 40]. (G) Uncropped western blot images depicting STAT5 and GAPDH. (H) Uncropped western blot images depicting AR, PSA, and GAPDH.
(PDF)

**S2 Fig. Western blots of nuclear localisation of STAT5 and STAT3.** (A) Uncropped western blot of STAT5, Lamin A/C, and GAPDH after cytoplasmic and nuclear fractionation. Abbreviation: M: MagicMark XP. (B+C) Uncropped western blot images depicting pSTAT3, STAT3, GAPDH and densitometric analysis of STAT3 relative to GAPDH.
(PDF)

**S3 Fig. Western blots of STAT5 activity in LNCaP-derived models after pimozide treatment.** (A) Uncropped western blot images depicting STAT5 and GAPDH and densitometric analysis of STAT5 relative to GAPDH. (B) Uncropped western blot of Lamin A/C, and GAPDH after cytoplasmic and nuclear fractionation. (C) Uncropped western blot of Lamin A/C, and STAT5 after cytoplasmic and nuclear fractionation and densitometric analysis of nuclear STAT5 relative to Lamin A/C. Abbreviations: M: MagicMark XP; WCL: whole cell lysate.
(PDF)

**S4 Fig. Western blots of STAT5 activity in LAPC4-derived models after pimozide treatment.** (A) Uncropped western blot images depicting STAT5 and GAPDH and densitometric analysis of STAT5 relative to GAPDH. (B) Uncropped western blot of Lamin A/C, and GAPDH after cytoplasmic and nuclear fractionation. (C) Uncropped western blot of Lamin A/C, and STAT5 after cytoplasmic and nuclear fractionation and densitometric analysis of nuclear STAT5 relative to Lamin A/C. Abbreviation: M: MagicMark XP.
(PDF)

**S5 Fig. Analysis of the relative STAT5 and AR activity after treatment with pimozide and enzalutamide.** (A) qPCR analysis of Cyclin D1 (CCND1) and BCL-xL (BCL2L1) in C4-2 and MR49F cells treated with 10 µM Pimozide for 8 h. (B) qPCR analysis of PSA/KLK3 in C4-2 cells and MR49F cells transfected with siCTRL, siSTAT5a, and siSTAT5b for 24 h.
(PDF)

**S6 Fig. Validation of STAT5a/b knockdown.** (A+B) qPCR analysis of STAT5a (A) and STAT5b (B) normalised to HPRT1 in C4-2 cells transfected with siCTRL, siSTAT5a, and siSTAT5b (all: final concentration 25 nM) for 48 h. (C+D) qPCR analysis of STAT5a (C) and STAT5b (D) in C4-2 cells transfected with siCTRL, siSTAT5a, and siSTAT5b (all: final concentration 25 nM) for 24 h, 48 h, and 72 h. (E) Western Blot of STAT5a/b and GAPDH in C4-2 cells after transfection with siCTRL, siSTAT5a, and siSTAT5b (all: final concentration 25 nM) for 48 h. (F) Western Blot of STAT5a/b and GAPDH in C4-2, MR49F, LAPC4-CTRL, LAPC4-EnzaR cells after transfection with siCTRL, siSTAT5a, and siSTAT5b (all: final concentration 25 nM) for 48 h. Abbreviation: M: MagicMark XP.
(PDF)

**S1 Table. Cell culture media for used cell lines.**
(DOCX)

**S2 Table. Antibodies and used dilutions.**
(DOCX)

**S1 Raw images.**
(PDF)

## Acknowledgments

The Department of Urology and Pediatric Urology, University Medical Center Mainz is acknowledged for its technical support. This work is part of Julia Bodenbender's MD thesis.

Besides, we would like to thank Andrea Lohse-Fischer, Ulrike Lotzkat, Jana Scholze, and Tiziana Siciliano of the Department of Urology, Technische Universität Dresden for their technical support during the revision process.

## Author Contributions

**Conceptualization:** Holger H. H. Erb.

**Funding acquisition:** Christian Thomas.

**Investigation:** Julia Bodenbender, Florian Handle, Tamara Diehl, Lukas Donix.

**Methodology:** Holger H. H. Erb, Julia Bodenbender.

**Software:** Florian Handle, Lukas Donix.

**Supervision:** Holger H. H. Erb, Eva Juengel, Zoran Culig, Christian Thomas.

**Writing – original draft:** Holger H. H. Erb.

**Writing – review & editing:** Holger H. H. Erb, Florian Handle, Igor Tsaur, Martin Gleave, Axel Haferkamp, Johannes Huber, Susanne Fuessel, Zoran Culig, Christian Thomas.

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
