## [Decision Letter · Decision Letter 0]

24 Apr 2020

PONE-D-20-06296

Assessment of STAT5 as a Potential Therapy Target in Enzalutamide-Resistant Prostate Cancer

PLOS ONE

Dear Professor Erb:,

Thank you for submitting your manuscript to PLOS ONE. After careful consideration, we feel that it has merit but does not fully meet PLOS ONE’s publication criteria as it currently stands. Therefore, we invite you to submit a revised version of the manuscript that addresses the points raised during the review process.

Please address the concerns raised by the reviewers. 

We would appreciate receiving your revised manuscript by May 24, 2020. To enhance the reproducibility of your results, we recommend that if applicable you deposit your laboratory protocols in protocols.io, where a protocol can be assigned its own identifier (DOI) such that it can be cited independently in the future. For instructions see: http://journals.plos.org/plosone/s/submission-guidelines#loc-laboratory-protocols

We look forward to receiving your revised manuscript.

Kind regards,

Daotai Nie, Ph.D.

Academic Editor

PLOS ONE

2. Please provide additional information about each of the cell lines used in this work, including any quality control testing procedures (authentication, characterisation, and mycoplasma testing). For more information, please see http://journals.plos.org/plosone/s/submission-guidelines#loc-cell-lines.

In addition, please provide the source, history and culture conditions of the LAPC4-CTRL, LAPC4-EnzaR, LNCaPabl-CTRL, and LNCaPabl-EnzaR cell lines used in this work.

3. In the Methods section, please provide the product number and nay lot numbers of the Enzalutamide and Pimozide purchased from Astellas Pharma and Sigma-Aldrich respectively for this study.

4. Please provide the product numbers of all antibodies used in the Western blot analysis in the Supplementary Table 1.

5. Please provide additional details about the methodology used for the Western blot analysis in the Methods section.

7. Please upload a copy of Supporting Information Table 7 which you refer to in your text on page 6.

Reviewers' comments:

Reviewer's Responses to Questions

**Comments to the Author**

1. Is the manuscript technically sound, and do the data support the conclusions?

Reviewer #1: Yes

Reviewer #2: Yes

Reviewer #3: Yes

2. Has the statistical analysis been performed appropriately and rigorously? 

Reviewer #1: Yes

Reviewer #2: Yes

Reviewer #3: Yes

3. Have the authors made all data underlying the findings in their manuscript fully available?

Reviewer #1: Yes

Reviewer #2: Yes

Reviewer #3: Yes

4. Is the manuscript presented in an intelligible fashion and written in standard English?

Reviewer #1: Yes

Reviewer #2: Yes

Reviewer #3: Yes

5. Review Comments to the Author

Reviewer #1: The submitted manuscript assessed the potential of STAT5 as a therapeutic target for enzalutamide resistant, metastatic, castration resistant prostate cancer (mCRPC). This study addresses an important and current problem for patients with mCRPC. The work is original and experiments are well designed, include appropriate controls, and supporting data. The manuscript is written in a clear, intelligent, and fair manner. The authors appropriately cite previous literature and consider their results in light of the current knowledge in the field. The conclusions are clearly stated and supported by the data. While most of the paper is very well written, there are numerous errors and inaccuracies. These are addressed below, and additional comments are provided.

1. The manuscript contains several typos, misplaced punctuation, and incorrect punctuation. The authors need to carefully review the entire manuscript, figures, and figure legends.

2. The STAT5 inhibition studies are critical for this paper and its conclusions. While pimozide has biologic and dose-dependent effects on both cell models in Figure 4 and table 1, the direct effects on STAT5 activity are less convincing and inconsistent in supplementary figures 4 and 5. A more direct and quantitative assay of STAT5 activity, such as phospho-STAT5 quantification or a STAT5 transcriptional reporter assay, is needed.

3. The effect of STAT5B knockdown in MR49F cells is important to the conclusions of this paper. The supplementary data includes validation of siRNA knockdown for C4-2, but not for MR49F or LAPC4. It is important to verify siRNA knockdown in all cells.

4. The authors need to carefully review the datasets used in the analysis and their presentation in the text and data. For example, the manuscript describes GSE58815, which does not appear to be a prostate dataset. The datasets are sometimes incorrectly labeled under the wrong samples, for example in Figure 3B which lists GSE5816 under C4-2, but this dataset appears to study LNCaP and not C4-2. Also, sometimes "GSE" is written as "GES"

5. The manuscript requires more detail about the ChIP-seq data analyses. How was the data normalized across samples. Was a negative control (IgG) considered in analyses?

6. The methods section includes description of cell cycle analysis, but the manuscript does not appear to include this data.

Reviewer #2: the authors do a good job of investigating the possibility of stat 5 as a target for treatment in enzalutamide resistance. THis is important work since there had been some evidence that this could be a target and even a potential mechanisms for resistance. With the caveat that they are restricted to cell line models, this effort seems to decrease any enthusiasm for stat 5 as a target. esp of interest is the possibility that the inhibitor used is likely to have off target effects .

of note they should probly shorten the conclusion for 4 double p spaced pages and add a brief caveat that the results are from cell lines that may or may not represent enza resistance disease

Reviewer #3: The manuscript addresses the need to better understand the role of STAT5 in enzalutamide-resistant prostate cancer and concluded that resistance mechanisms are complex and that STAT5 may not be integral to this process. A concern with the manuscript is the lack of clarity in the introduction as to the specific question being addressed by the authors. Just asking whether STAT5 has role in drug resistance is too general a question. A more focused approach could improve the manuscript. For example, the first figures showed that STAT5b expression was heterogenous in resistant cells but not in sensitive cells. Examining the molecular events behind this observation could make a better manuscript. Additional comments follow.

1) The statement that “STAT5b is highly heterogenous in enzalutamide-resistant xenografts while expression in enzalutamide-sensitive CRPC xenografts is rather consistent “ should be supported by statistical information. The descriptive terms used are rather vague.

2) While it is understood that approach combining bioinformatics (Fig. 1, 3) and wet bench experiments (Fig. 2,4,5) necessitates the use of different cell lines, the results from each of the different cell lines used is unclear. It is suggested that the authors more clearly summarize trends per cell line either in each section or at the end. There is a concern that cell-line specific effects may be overlooked. This is especially important for the isobologram analysis as well.

6. PLOS authors have the option to publish the peer review history of their article (what does this mean?). If published, this will include your full peer review and any attached files.

Reviewer #1: No

Reviewer #2: No

Reviewer #3: No

---

## [Author Response · Author response to Decision Letter 0]

1 Jun 2020

On the half of all authors, we would like to take this opportunity to express our sincere gratitude to the reviewers who identified areas of our manuscript that needed correction or modification.

Below you find the detailed response to the reviewers’ comments:

The formatting of the manuscript is now following the journals guidelines.

2. Please provide additional information about each of the cell lines used in this work, including any quality control testing procedures (authentication, characterisation, and mycoplasma testing). For more information, please see http://journals.plos.org/plosone/s/submission-guidelines#loc-cell-lines.

In addition, please provide the source, history and culture conditions of the LAPC4-CTRL, LAPC4-EnzaR, LNCaPabl-CTRL, and LNCaPabl-EnzaR cell lines used in this work.

The additional information has been added to S1 Table. 

3. In the Methods section, please provide the product number and nay lot numbers of the Enzalutamide and Pimozide purchased from Astellas Pharma and Sigma-Aldrich respectively for this study.

The missing information has been added to the materials and method section of the manuscript

4. Please provide the product numbers of all antibodies used in the Western blot analysis in the Supplementary Table 1.

The additional information has been added to S2 Table. 

5. Please provide additional details about the methodology used for the Western blot analysis in the Methods section.

Additional information to the western blot methodology has been added. 

6. PLOS ONE now requires that authors provide the original uncropped and unadjusted images underlying all blot or gel results reported in a submission's figures or Supporting Information files. This policy and the journal's other requirements for blot/gel reporting and figure preparation are described in detail at https://journals.plos.org/plosone/s/figures#loc-blot-and-gel-reporting-requirements and https://journals.plos.org/plosone/s/figures#loc-preparing-figures-from-image-files. When you submit your revised manuscript, please ensure that your figures adhere fully to these guidelines and provide the original underlying images for all blot or gel data reported in your submission. See the following link for instructions on providing the original image data: https://journals.plos.org/plosone/s/figures#loc-original-images-for-blots-and-gels.

In your cover letter, please note whether your blot/gel image data are in Supporting information or posted at a public data repository, provide the repository URL if relevant, and provide specific details as to which raw blot/gel images, if any, are not available. Email us at plosone@plos.org if you have any questions.

The raw western blot pictures are now displayed in the file S1_raw_pictures.pdf. 

7. Please upload a copy of Supporting Information Table 7 which you refer to in your text on page 6.

Due to rearrangement of the figures during the writing process of the manuscript the supporting information has been rearranged and the numbering has not been corrected in the manuscript. This issue is now fixed. 

Reviewer #1: The submitted manuscript assessed the potential of STAT5 as a therapeutic target for enzalutamide resistant, metastatic, castration resistant prostate cancer (mCRPC). This study addresses an important and current problem for patients with mCRPC. The work is original and experiments are well designed, include appropriate controls, and supporting data. The manuscript is written in a clear, intelligent, and fair manner. The authors appropriately cite previous literature and consider their results in light of the current knowledge in the field. The conclusions are clearly stated and supported by the data. While most of the paper is very well written, there are numerous errors and inaccuracies. These are addressed below, and additional comments are provided.

1. The manuscript contains several typos, misplaced punctuation, and incorrect punctuation. The authors need to carefully review the entire manuscript, figures, and figure legends.

The manuscript has been reviewed carefully by the authors, and all mistakes have been corrected.

2. The STAT5 inhibition studies are critical for this paper and its conclusions. While pimozide has biologic and dose-dependent effects on both cell models in Figure 4 and table 1, the direct effects on STAT5 activity are less convincing and inconsistent in supplementary figures 4 and 5. A more direct and quantitative assay of STAT5 activity, such as phospho-STAT5 quantification or a STAT5 transcriptional reporter assay, is needed.

We want to thank the reviewer for his comment. Detecting pSTAT5 is highly challenging and a very good controlled IP is necessary to detect reproducible results. This has been shown in several studies by Prof. Nevalainen the leading STAT5 expert in PCa. However, we tested the antibody Phospho-Stat5 (Tyr694) (D47E7) XP® Rabbit mAb #4322 (Cell Signaling). Sadly, no signal could be detected (Figure 1). 

Figure 1: Exemplary Western Blot for pSTAT5

Therefore, we decided to follow a different approach. As STAT5 has to be phosphorylated before dimerisation and nuclear transport, we decided to show nuclear STAT5 levels as a surrogate for STAT5 activity. Similar evaluation was performed by Udhane et al. to determine STAT5 activity in tissue [1]. To undermine our localization data we also added the regulation of two genes regulated by a STAT5 response element. This approach was chosen, as reporter genes assays also just represent the changes in expression of a luciferase gene regulated by a STAT5 response element. Therefore, we believe that a decrease of STAT5 localization and the decrease in STAT5 target gene expression is sufficient to show a decrease in STAT5 activity after pimozide treatment.

3. The effect of STAT5B knockdown in MR49F cells is important to the conclusions of this paper. The supplementary data includes validation of siRNA knockdown for C4-2, but not for MR49F or LAPC4. It is important to verify siRNA knockdown in all cells.

The authors agree with the reviewer's criticism and the missing data has been added (S3 Fig F) 

4. The authors need to carefully review the datasets used in the analysis and their presentation in the text and data. For example, the manuscript describes GSE58815, which does not appear to be a prostate dataset. The datasets are sometimes incorrectly labeled under the wrong samples, for example in Figure 3B which lists GSE5816 under C4-2, but this dataset appears to study LNCaP and not C4-2. Also, sometimes "GSE" is written as "GES"

We would like to thank the reviewer for pointing out this issue. We carefully reviewed the databases and corrected the issue. 

5. The manuscript requires more detail about the ChIP-seq data analyses. How was the data normalized across samples. Was a negative control (IgG) considered in analyses?

The ChIP-seq analysis was based on the only publicly available AR ChIP-seq dataset that contained modulation of AR activity and replicate samples (GSE65066). Sadly, this dataset did not provide IgG control. Therefore, we included a second high quality dataset (with IgG control, GSE62442) that originated from AR ChIP-seq on a variety of cell lines, including LNCaP cells (= the parental cell line of C4-2 cells) and used the annotated androgen-binding regions from this dataset to look at the AR binding intensity in C4-2 cells.The reported values are the scaled output of MACS2 as deposited in the GEO database by the original authors. Due to the limitations of this dataset we considered the results to be only qualitative and did not perform further quantitative statistical analysis of this dataset. Our conclusion from the ChIPseq data was that the AR is unlikely to affect STAT5 expression directly. To test this, we performed the biological experiment shown in Fig. 3E, which proved our point that short term treatment with androgens and enzalutamide does not affect STAT5 expression. To address the reviewer's concern, we have included additional information in the figure legend and the materials and methods section.

6. The methods section includes description of cell cycle analysis, but the manuscript does not appear to include this data.

In a preceding version of the manuscript, cell cycle analysis was included. After a constructive discussion with the authors, the cell cycle analysis was removed due to its off-topic character. However, the material and methods section has not been changed. The description has now been removed. 

Reviewer #2: the authors do a good job of investigating the possibility of stat 5 as a target for treatment in enzalutamide resistance. THis is important work since there had been some evidence that this could be a target and even a potential mechanisms for resistance. With the caveat that they are restricted to cell line models, this effort seems to decrease any enthusiasm for stat 5 as a target. esp of interest is the possibility that the inhibitor used is likely to have off target effects .

of note they should probly shorten the conclusion for 4 double p spaced pages and add a brief caveat that the results are from cell lines that may or may not represent enza resistance disease

The authors would like to thank the reviewer for his time and his constructive criticism. We tried to shorten the conclusion and added the information that the study is based only on in vitro experiments.

Reviewer #3: The manuscript addresses the need to better understand the role of STAT5 in enzalutamide-resistant prostate cancer and concluded that resistance mechanisms are complex and that STAT5 may not be integral to this process. A concern with the manuscript is the lack of clarity in the introduction as to the specific question being addressed by the authors. Just asking whether STAT5 has role in drug resistance is too general a question. A more focused approach could improve the manuscript. For example, the first figures showed that STAT5b expression was heterogenous in resistant cells but not in sensitive cells. Examining the molecular events behind this observation could make a better manuscript. Additional comments follow.

We would like to thank the reviewer for his time and effort. In our manuscript, we focused on the role of STAT5 in enzalutamide resistant models. We tried to sharpen our aim in the introduction. The mechanisms how STAT5 is involved in the process of developing enzalutamide resistance and the regulation of STAT5 in this process has been published by Udhane et al. (doi 10.1158/1535-7163.MCT-19-0508, published January 2020). Moreover, in further studies of our laboratory the mechanisms will be investigated in more detail. 

1) The statement that "STAT5b is highly heterogenous in enzalutamide-resistant xenografts while expression in enzalutamide-sensitive CRPC xenografts is rather consistent "should be supported by statistical information. The descriptive terms used are rather vague.

We would like to thank the reviewer for his criticism. We added descriptive values to the manuscript 

2) While it is understood that approach combining bioinformatics (Fig. 1, 3) and wet bench experiments (Fig. 2,4,5) necessitates the use of different cell lines, the results from each of the different cell lines used is unclear. It is suggested that the authors more clearly summarize trends per cell line either in each section or at the end. There is a concern that cell-line specific effects may be overlooked. This is especially important for the isobologram analysis as well.

The authors agree that a sum up at the end of each result section is helpful to increase clarity. An additional sum up has been added after the section “Influence of STAT5a- and STAT5b-knockdown on cell viability in the presence and absence of enzalutamide”. As there is already a sum up after the isobologram section, no additional sum up has been added. 

1. Udhane V, Maranto C, Hoang DT, Gu L, Erickson A, Devi S, et al. Enzalutamide Induced Feed-Forward Signaling Loop Promotes Therapy-Resistant Prostate Cancer Growth Providing an Exploitable Molecular Target for Jak2 Inhibitors. Molecular cancer therapeutics. 2019. doi: 10.1158/1535-7163.MCT-19-0508. PubMed PMID: 31548294.

---

## [Decision Letter · Decision Letter 1]

23 Jul 2020

Assessment of STAT5 as a Potential Therapy Target in Enzalutamide-Resistant Prostate Cancer

PONE-D-20-06296R1

Dear Dr. Erb,

We’re pleased to inform you that your manuscript has been judged scientifically suitable for publication and will be formally accepted for publication once it meets all outstanding technical requirements.

Kind regards,

Daotai Nie, Ph.D.

Academic Editor

PLOS ONE

Additional Editor Comments (optional):

Reviewers' comments:

Reviewer's Responses to Questions

**Comments to the Author**

1. If the authors have adequately addressed your comments raised in a previous round of review and you feel that this manuscript is now acceptable for publication, you may indicate that here to bypass the “Comments to the Author” section, enter your conflict of interest statement in the “Confidential to Editor” section, and submit your "Accept" recommendation.

Reviewer #1: All comments have been addressed

Reviewer #2: All comments have been addressed

Reviewer #3: All comments have been addressed

2. Is the manuscript technically sound, and do the data support the conclusions?

Reviewer #1: Yes

Reviewer #2: Yes

Reviewer #3: Yes

3. Has the statistical analysis been performed appropriately and rigorously? 

Reviewer #1: Yes

Reviewer #2: Yes

Reviewer #3: Yes

4. Have the authors made all data underlying the findings in their manuscript fully available?

Reviewer #1: Yes

Reviewer #2: Yes

Reviewer #3: Yes

5. Is the manuscript presented in an intelligible fashion and written in standard English?

Reviewer #1: Yes

Reviewer #2: Yes

Reviewer #3: Yes

6. Review Comments to the Author

Reviewer #1: (No Response)

Reviewer #2: my concerns have been addressed , the conclusion has been shortened and the manuscript addresses the problem with cell line data

Reviewer #3: (No Response)

7. PLOS authors have the option to publish the peer review history of their article (what does this mean?). If published, this will include your full peer review and any attached files.

Reviewer #1: No

Reviewer #2: **Yes: **Glenn Bubley

Reviewer #3: No